# Atrazine Removal from Municipal Wastewater Using a Membrane Bioreactor

**DOI:** 10.3390/ijerph17072567

**Published:** 2020-04-09

**Authors:** Mohanad Kamaz, Steven M. Jones, Xianghong Qian, Michael J. Watts, Wen Zhang, S. Ranil Wickramasinghe

**Affiliations:** 1Ralph E. Martin Department of Chemical Engineering, University of Arkansas, Fayetteville, AR 72701, USA; makamaz@uark.edu; 2Ministry of Oil, Gas Filling Company (CFO), Baghdad 10011, Iraq; 3Garver, 5251 DTC Parkway, Suite 420, Greenwood Village, CO 80111, USA; SMJones@GarverUSA.com; 4Department of Biomedical Engineering, University of Arkansas, Fayetteville, AR 72701, USA; xqian@uark.edu; 5Garver, 14160 N Dallas Parkway, Suite 850, Dallas, TX 75254, USA; MJWatts@GarverUSA.com; 6Department of Civil Engineering, University of Arkansas, Fayetteville, AR 72701, USA; wenzhang@uark.edu

**Keywords:** aerobic, anoxic, biodegradation, direct potable reuse, genetic profiling

## Abstract

As the demand for potable water increases, direct potable reuse of wastewater is an attractive alternative method to produce potable water. However, implementation of such a process will require the removal of emerging contaminants which could accumulate in the drinking water supply. Here, the removal of atrazine, a commonly used herbicide, has been investigated. Using real and synthetic wastewater, as well as sludge from two wastewater treatment facilities in the United States in Norman, Oklahoma and Fayetteville, Arkansas, atrazine removal has been investigated. Our results indicate that about 20% of the atrazine is removed by adsorption onto the particulate matter present. Significant biodegradation of atrazine was only observed under aerobic conditions for sludge from Norman, Oklahoma. Next-generation sequencing of the activated sludge revealed the abundance of *Noncardiac* with known atrazine degradation pathways in the Norman aerobic sludge, which is believed to be responsible for atrazine biodegradation in our study. The detection of these bacteria could also be used to determine the likelihood of biodegradation of atrazine for a given wastewater treatment facility.

## 1. Introduction

Water is a very valuable natural resource. The rising world population and urbanization has led not only to an increased need for drinking water, but also large quantities of discharged wastewater. Consequently, recycle and reuse of wastewater is of growing importance [1,2]. Treated wastewater can enter reservoirs which provide raw water feed to a water treatment facility. The treated water is then distributed through the potable water distribution system to residents. As the time between discharge from the wastewater treatment plant and introduction to the potable water distribution system decreases, there is an increasing risk that emerging contaminants, such as personal care products, pharmaceuticals, pesticides, industrial chemicals, and fire retardants will accumulate in the drinking water. Wastewater treatment facilities are not designed to remove these emerging micropollutants, which are frequently unregulated. These pollutants eventually lead to environmental degradation. Consequently, there is an urgent need to determine the fate of these contaminants and develop regulatory limits in the drinking water supply. In particular, it is important to determine if adequate removal is provided by the wastewater treatment process at a given facility.

Direct potable reuse of wastewater is a process where the treated wastewater is directly introduced into the potable water distribution system after appropriate treatment and monitoring. Currently, direct potable reuse is used in few locations, the most well-known examples being Windhoek, Namibia; Big Springs, TX, USA; Cloudcroft, NM, USA; and Fountain Valley, CA, USA. However, demand is increasing. Currently, the El Paso Water Utilities, El Paso, TX, USA, is designing the world’s largest facility with a capacity of 37,800 m^3^/day (10 MGD).

We have investigated the removal of micropollutants that affect the human endocrine system, known as endocrine disrupting compounds [3]. Here, we focus on atrazine or 2-chloro-4-ethylamino-6-isopropylamino-s-triazine, C_8_H_14_ClN_5_. Atrazine is used as a herbicide to protect many crops, including corn, sorghum, tea, sugarcane, and various fruit crops [4]. In fact, atrazine is one of the most widely used herbicides in the world [5]. It is a carcinogen that can disrupt the endocrine system of frogs at concentrations detected in the environment [6]. In the case of atrazine, the World Health Organization (WHO) has indicated a guideline of 2 μg L^−1^. The WHO Standard and US EPA Primary Drinking Water Standard for atrazine are equivalent (on mg/L basis). In surface waters and wastewater treatment plant effluents in the US, concentrations of 32–790 ng L^−1^ and 49–870 ng L^−1^ have been detected [7,8]. In our previous work we investigated the removal of five model micropollutants using a laboratory-scale membrane bioreactor (MBR) [9]. However, in agreement with previous studies, we found that atrazine removal was limited.

Several investigators have investigated atrazine removal during different wastewater treatment processes. In an early study, Ho et al. [10] investigated the effectiveness of sand filtration and adsorption by granular activated carbon. They found that sand filtration was ineffective; however, activated carbon was successful in adsorbing atrazine. Buttiglieri et al. [4] investigated the removal of atrazine using a semi-synthetic feed in a MBR pilot plant. Removal was low, being less than 25%, mainly by adsorption. In an interesting study, Nguyen et al. [11] investigated the removal of atrazine in a MBR containing a mixed culture of bacteria and white-rot fungi. Their result indicated that atrazine and other emerging organic micropollutants, that are resistant to bacterial degradation, are successfully degraded by white-rot fungal enzyme, laccase, combined with a redox mediator such as 1-hydroxy benzotriazole.

Mai et al. [12,13] investigated the use of nitrifying trickling filters for the removal of atrazine. In general, moderate (50%) removal of atrazine was observed. They indicated that biodegradation was the main process for atrazine removal. These investigators noted that the addition of organic matter can promote microorganism growth, and hence, removal by adsorption. However, the addition of sucrose had a small improvement in atrazine removal, suggesting limited removal by biodegradation.

Recently, several investigators have considered slight modifications to existing unit operations in order to enhance the removal of micropollutants. Wei and Hoppe-Jones [14] investigated the use of an anoxic MBR system with combined nanofiltration for the removal of emerging micropollutants. Atrazine removal was low, being less than 40%. Borea et al. [15] considered the use of an electrochemical MBR. A significant improvement in atrazine removal was observed. Atrazine is a weak base with a pKa value of 1.7—thus, atrazine is neutral in wastewater. These investigators suggest that in the electrochemical MBR, adsorption of atrazine onto electrically generated coagulants and flocs explains the higher removal.

In recent studies, Derakhshan et al. [16,17] obtained very high levels of atrazine removal using a membrane photobioreactor and a moving bed biofilm reactor. In the former case, atrazine removal was due to a symbiotic relationship between algae and bacteria that led to co-metabolism. In the latter case, these investigators found that increasing salinity influenced biodegradation. Mukherjee et al. [18] developed a ceramic ultrafiltration membrane for use in a MBR for atrazine removal. More than 90% of atrazine removal was obtained for simulated wastewaters due to a combination of biodegradation and adsorption onto the ceramic membrane.

The main removal pathways for emerging micropollutants are adsorption, biodegradation, and volatilization [19]. In the case of atrazine, volatilization is insignificant. While biodegradation of atrazine is possible, previous studies are scattered, indicating a wide variation in the level of biodegradation. High levels of atrazine removal have been reported using genetically modified organisms in an MBR [20]. Evidence of biodegradation of atrazine is provided by Jones et al. [21] who detected the presence of atrazine metabolites in wastewater.

In our previous work [22], we developed a laboratory-scale MBR unit that contains both anoxic and aerobic tanks. The aerobically treated water was recirculated back to the anoxic tank to mimic the actual operation. Here, we use this system to investigate atrazine removal in a laboratory-scale MBR. Actual sludge was obtained from the City of Norman, OK, USA, Water Reclamation Facility and the West Side Wastewater Treatment Facility in Fayetteville, AR, USA. Synthetic and real wastewater from the West Side Wastewater Treatment Facility in Fayetteville, AR, USA was used to feed the MBR. The atrazine concentration was measured throughout the treatment process, and the next-generation sequencing of the activated sludge from the MBR was also conducted to help explain the performance difference in atrazine removal.

## 2. Materials & Methods

Atrazine (≥97%) was purchased from Tokyo Chemical Industry (TCI) (Chuo-ku, Tokyo). Liquid phenol, sodium nitroprusside dihydrate (≥98%), sodium hydroxide, sodium hypochlorite, sodium bicarbonate, ammonium acetate, magnesium sulphate, calcium chloride, ammonium nitrate, and ethanol were purchased from VWR (Radnor, PA, USA) and used as received with no further purification. Monopotassium phosphate, dipotassium phosphate, zinc sulfate, manganese sulfate, magnesium sulfate, ferric chloride, and yeast extract were purchased from Sigma-Aldrich (St Louis, MO, USA). HPLC-grade acetonitrile from EMD Millipore (Bedford, MA, USA) and de-ionized (DI) water (Milli-Q, 18.2 MΩ cm) were employed as the mobile phase for HPLC analysis. Chemical oxygen demand (COD) kits were purchased from CHEMetrics (Midland, VA, USA) with a range of 0–1500 ppm. Total ammonia nitrogen (TAN) was determined using an established method [23] with ±0.02 mg/L accuracy. Nitrate nitrogen (NO_3_-N) detection kits were purchased from HACH (Loveland, CO, USA).

### 2.1. Atrazine Spiking

Atrazine was detected at 0.02 ppm levels in the wastewater obtained from the treatment facility in Fayetteville. The presence of atrazine in the wastewater at the Norman treatment facility has also been reported [24,25]. However, given the variation in the levels of atrazine that are detected, in this study, all wastewater streams were spiked with atrazine. Atrazine was stored at room temperature. Atrazine was dissolved in a ethanol/water mixture 1:1 *v*/*v*, followed by sonication prior to spiking into the wastewater.

### 2.2. Atrazine Detection

High-Performance Liquid Chromatography 1260 Infinity HPLC from Agilent Technologies (Santa Clara, CA, USA) operated in reverse phase mode was used to quantify the concentration of atrazine, as described in our earlier work [22]. A Luna C_18_ column from Phenomenex, 5 μm, size 250 × 4.6 mm (Torrance, CA, USA) was used. Briefly, the mobile phase consisted of acetonitrile and DI water at a flow rate of 0.75 mL/min, with a linear gradient varying from 10 to 100% acetonitrile. A diode array detector (DAD) was used for detection. The limit of detection was 5 ppb at a wavelength of 222 nm.

### 2.3. Membrane Bioreactor

Figure 1 is a schematic diagram of the laboratory-scale MBR. The MBR consisted of two 35 L tanks—one operated under anoxic conditions, and the other under aerobic conditions. Polyvinylidene difluoride (PVDF) flat sheet microfiltration membranes provided by the Lantian Corporation (Lantian Inc., Wuxi, China) were used in this study. The nominal membrane pore size was 0.08 μm and the effective surface area was 0.1102 m^2^. A sparger was used to provide continuous air bubbling to the aerobic tank, while the contents of the anoxic tank were homogenized using a mechanical mixer.

Wastewater was collected from the West Side Wastewater Treatment Facility (Fayetteville, AR, USA), while synthetic wastewater (recipe provided in Table 1) was prepared and used for batch experiments. The wastewater contained mainly dissolved organic matter and nutrients (carbon, nitrogen and phosphorus). Anoxic and aerobic sludge was collected from Norman, OK, USA and Fayetteville, AR, USA and immediately seeded into our laboratory MBR.

After the sludge was loaded into the MBR and fed with wastewater, it was allowed to acclimatize for up to 2–3 weeks until consistent nutrient removal, as indicated by COD, TAN, and NO_3_-N, was achieved. Dissolved oxygen (DO) levels in the anoxic and aerobic tanks were monitored to ensure the existence of two redox zones, which promote nitrification (aerobic) and denitrification (anoxic). The DO level in the anoxic tank was kept below 1 ppm, whereas it was above 2 ppm in the aerobic tank depending on the (mixed liquor suspended solids) MLSS circulation rate and total suspended solids (TSS) concentration.

Samples from the aerobic and anoxic tanks were collected at various times and COD, TAN, NO_3_-N, and total suspended solids (TSS) determined. The DO was continuously measured using a SympHony^TM^ dissolved oxygen probe VWR International (Radnor, PA, USA). COD and NO_3_-N were determined according to the protocols given by the manufacturers of the kits purchased from HACH Company (Loveland, CO, USA). The laboratory-scale MBR used here was run in semi-continuous mode. Treated water was removed intermittently every 12 h (or 20 h for longer run times) as permeate after microfiltration. The permeate flux was about 50 Lm^−2^hr^−2^. Wastewater was added to replace the treated water that was removed. In order to maintain a solids retention time (SRT) of 35 days, a fraction of the sludge from both tanks was wasted each week.

Upon achieving the targeted levels of COD (<30 ppm), TAN (<5 ppm), and NO_3_-N (<5 ppm), the system was run for a further 2–3 weeks, after which atrazine was added to the aerobic and anoxic tanks. First, atrazine was spiked into two 10 L volumes of wastewater. These two 10 L volumes of wastewater were then added to the anoxic and aerobic tanks, each of which contained 17 L of MLSS each (either from the Fayetteville or Norman Treatment facility). The final spiked atrazine concentration in the aerobic and anoxic tanks, each of which had a total volume of 27 L, was 1 ppm.

Within 5 min of spiking the two 10 L volumes of wastewater, a 50 mL sample of the wastewater was removed to determine the atrazine concentration. After the addition of the 10 L volumes of the wastewater to the anoxic and aerobic tanks, a second sample was withdrawn again within 5 min in order to determine the atrazine concentration. This is the atrazine concertation at zero time. Further samples were collected after 4, 8, and 12 h of operation. The concentration of the atrazine in the filtrate was determined. MLSS was recirculated between the tanks at 32 mL/min using a peristaltic pump. The TSS concentrations were 6024 and 7029 ppm in the anoxic and aerobic tanks for sludge from Norman, OK, USA and 4535 and 7244 ppm for sludge obtained from Fayetteville, AR, USA.

### 2.4. Batch Experiments for Atrazine Degradation

Additional batch experiments were conducted using synthetic wastewater where there is no recirculation of the MLSS between the anoxic and aerobic tanks in order to determine which conditions are responsible for atrazine degradation. Two 3 L tanks were used. The aerobic tank was continuously aerated, while the anoxic tank had a mechanical mixer and no aeration. The initial MLSS concentrations were about 6024 and 7029 mg/L in the anoxic and aerobic tanks for sludge from Norman, OK, USA and 4920 and 5958 mg/L for sludge obtained from Fayetteville, AR, USA.

Synthetic wastewater was used in order to avoid the natural variation in the composition of real wastewater. Table 1 gives the components that were added to DI water to make synthetic wastewater. The synthetic wastewater (1 L) was spiked with atrazine (2.5 mg) and added to the two 3 L tanks in order to bring the MLSS volume to 2.5 L and the atrazine concentration to 1 ppm. Samples were removed from the anoxic and aerobic tanks (10 mL) right after adding the spiked wastewater referred to as 0 h, and after 8, 20, 30, and 45 h. The samples were analyzed by HPLC in order to determine the atrazine concentration.

### 2.5. Microbial Degradation

Experiments were also conducted in order to determine the ability of the microorganisms present in the Fayetteville and Norman sludges to use atrazine as a carbon source. After settling the MLSS from our MBR, an aliquot of 1 mL activated sludge was transferred to 15 mL centrifuge tubes. Bushnell Haas Broth (BH broth) 9 mL [26] containing atrazine was added to the solution to bring its concentration to 30 ppm. BH broth would supply nitrogen and the required nutrients for the growth of the microorganisms. After depletion of the initial carbon present in the activated sludge, the only source of carbon is atrazine. Table 2 indicates the compositions of the BH broth.

The tubes were incubated at 37 °C with shaking at 150 rpm. The disappearance of the atrazine from the supernatant was monitored by collecting samples for HPLC. Control samples with BH buffer and standards without microorganisms were also incubated at the same conditions to monitor the changes in atrazine concentration.

### 2.6. Next-Generation Sequencing

From the aerobic and anoxic tanks, 2 mL of MLSS was extracted and used for DNA extraction. The samples were taken at the end of the run. DNA was extracted with a soil DNA extraction kit (DNeasy PowerSoil Kit, Qiagen, Germantown, MD, USA). The protocol recommended by the manufacturer was followed. The DNA was stored at −20 °C prior to sequencing analysis.

Following extraction, the DNA concentration was measured with a Take3 micro-volume plate on a BioTek Synergy H1 microplate reader (Winooski, VT, USA). The DNA was later submitted for next-generation sequencing at CD Genomics (Shirley, NY, USA). The sequencing followed the protocol described in Ul-Hasan et al. [27]. Briefly, the samples were processed with Illumina MiSeq PE300 platform (Illumina, San Diego, CA, USA) to analyze the V4-V6 region for the 16S rRNA of bacteria and archaea (forward primer 515F 5′- GTGYCAGCMGCCGCGGTAA-3′, and reverse primer 926R 5′-CCGYCAATTYMTTTRAGTTT-3′). Raw sequences were clustered into Operational Taxonomic Units (OTUs) using QIIME2 pipelines for sequence analyses. Shannon’s and Simpson’s diversity indices were calculated for diversity, and phylogenetic groups (i.e., phyla, class, order) were used to compile taxa for richness. All statistical tests and visualizations were performed in R [27].

## 3. Results and Discussion

Table 3 gives the range of the wastewater quality parameters for the sewage collected after primary sedimentation from the West Side Wastewater Treatment Facility in Fayetteville. Depending on the time of year, there can be significant variation in water quality. After loading the aerobic and anoxic tanks, the MBR was run for 2–3 weeks in order to acclimatize microorganisms based on achieving the targeted levels of COD, TAN, and NO_3_-N (Section 2.3). Approximately 12 L of make-up wastewater were added every 12 h. Prior to the addition of the makeup wastewater, the target levels of the main water quality parameters were: COD (<30 ppm), TAN (<5 ppm), and NO_3_-N (<5 ppm). These target levels were reached after 2–3 weeks of operation. The MLSS in all runs was below 10,000 mg/L, which is an ideal range for efficient full-scale MBR operation [28]. The COD in the permeate from the membrane was around 50% lower, relative to the feed [22].

Next, the aerobic and anoxic tanks were spiked with atrazine. Figure 2, Figure 3 and Figure 4 give the percentage removal of COD, TAN, and NO_3_-N after spiking with atrazine over a 12 h period. These results are consistent with our previous studies and indicate the stable operation of the MBR [22]. As the MLSS was recirculated between the aerobic and anoxic tanks, Figure 4 indicates a gradual removal of nitrate in the aerobic tank.

Figure 5 and Figure 6 give the percentage atrazine removal in MBRs using the Fayetteville and Norman sludges, respectively. Error bars show the range of values for three repeat experiments. As can be seen, the maximum atrazine removal using Fayetteville sludge is about 20%. This level of removal was reached rapidly, and there is little change over the entire run. In our previous work [22] we showed that most of this removal was by adsorption with minimal biodegradation. Figure 6 indicates that much better removal is obtained using the Norman sludge. The percentage removal in the aerobic tank is higher than the anoxic tank. After 12 h of operation, about 60% of removal was achieved in the aerobic tank. Given that the percentage removal increases with run time, the MBR was run for 20 h, leading to almost 85% removal in the aerobic tank.

The effectiveness of the two sludges for the biodegradation of atrazine was further investigated in batch experiments where there was no mixing of the MLSS. Biodegradation under aerobic and anoxic conditions was studied. Figure 7 gives the results for Fayetteville sludge, while Figure 8 gives results for the Norman sludge. Figure 7 indicates that little biodegradation occurred in either of the anoxic and aerobic tanks for the Fayetteville sludge.

Figure 8 shows different removal efficiency for atrazine by the Norman sludge. About 20% of removal is obtained in the anoxic tank with little change over time. This percentage removal is similar to the percentage of removal by adsorption for MBR operation using Fayetteville sludge [22]. However, in the aerobic tank, significant biodegradation occurs. After 30 h, 100% of the atrazine removal is observed. Thus, the batch experiments suggest that under aerobic conditions, the Norman sludge can effectively biodegrade atrazine.

Finally, additional experiments were conducted to determine if the microorganism can utilize atrazine as a carbon source. Experiments were only conducted under aerobic conditions for both sludges, as minimal biodegradation was observed under anoxic conditions. The results are shown in Figure 9, where the experiments were run for 140 h. Upon depletion of the initial carbon sources in the sludge, the only carbon source present is atrazine. The initial atrazine concentration was 30 ppm.

Figure 9 indicates little change in the atrazine concentration for the Fayetteville sludge over 140 h. The small initial decrease in atrazine concentration could be mainly due to adsorption. However, in the Norman sludge, after the small initial decrease in atrazine concentration due to the adsorption of atrazine, a continued decrease in atrazine concentration was observed over time. After about 80 h of incubation, the rate of atrazine degradation increases, suggesting that few other carbon sources are available to the microorganisms.

Our results suggest that there are microorganisms in the aerobic sludge from Norman which are able to biodegrade atrazine. Our results indicate little or no rejection of atrazine by the microfiltration membrane, which is not unexpected given the large pore size relative to the molecular weight of atrazine. In earlier studies, Kamaz et al. [22] indicated that about 20% of atrazine removal was by adsorption, highlighting the fact that little biodegradation occurs with the Fayetteville sludge. The laboratory-scale MBR used here was run in semi-continuous mode. By using this mode of operation, we were able to compare sludges from two different locations under controlled conditions.

In order to explore this further, we conducted next-generation sequencing on the activated sludge from both the aerobic and anoxic tanks of the MBR for both Norman and Fayetteville sludge. Samples were taken for analysis at the end of each run, 12 h for the Fayetteville sludge and 20 h for the Norman sludge. After the quality control filtering process on the obtained splicing sequences, there were 13,708 to 27,754 high-quality clean tags in the four sludge samples, including Fayetteville–aerobic (Fay.AE), Fayetteville–anoxic (Fay.AN), Norman–aerobic (Nor.AE), and Norman–anoxic (Nor.AN). Microbial diversity was assessed within the community (alpha diversity) and between the collection of samples (beta diversity). Ace and Chao1 indices show the Fayetteville sludges have more species than Norman sludges, while the Shannon and Simpson indices show that Fayetteville aerobic sludge has the highest richness and evenness, followed by Norman anaerobic sludge.

The operational taxonomic units (OTU) were clustered, and the relative abundance of the annotated tags in each taxonomy levels were calculated. In all samples tested, Actinobacteria are the dominant phylum, followed by Proteobacteria, which is consistent with other wastewater studies [29]. Figure 10 shows the taxonomy distribution at the family level for all four sludge samples. Various percentages of *Nocardiaceae* were found within the four samples, including less than 1% in both Fayetteville sludges, 15.3% in Norman aerobic sludge, and 7.4% in Norman anoxic sludge. They are a family of aerobic, gram-positive actinomycetes that are commonly found in soil and water. Specifically, atrazine degradation was documented with the help of *Nocardioides* sp. [30]. In addition, atrazine metabolism by Norcardia was further elucidated for its initial pathway and synthesis of potential metabolites [31,32]. The presence of this family could help explain the effective biodegradation in Norman sludge, but not Fayetteville sludge.

Our future work will focus on running a MBR in continuous mode onsite, using a side stream from the wastewater treatment facility. The MBR will be run for several months to ensure steady state operation. Further, it will be important to follow changes in the microbial communities over time. Next-generation sequencing will be conducted at regular intervals over several months of operation in order to relate changes in the microbial communities to changes in performance and atrazine removal.

## 4. Conclusions

Validating the removal of atrazine is difficult. In this study, we established a laboratory-scale MBR containing aerobic and anoxic tanks. Atrazine removal was investigated using real wastewater from the West Side Wastewater Treatment Facility, Fayetteville, AR, USA. Sludge was obtained from the West Side Wastewater Treatment Facility, Fayetteville, AR, USA and the City of Norman, OK, USA, Water Reclamation Facility. Up to 20% of the atrazine was removed by adsorption onto particulate matter present in the wastewater.

Little biodegradation of atrazine was observed for the Fayetteville sludge. However, for the sludge from Norman, OK, USA, significant biodegradation was observed under aerobic conditions. In order to further confirm this observation, we conducted next-generation sequencing on the activated sludge from both aerobic and anoxic tanks of the MBR for the sludges from both locations. Varying amounts of *Nocardiaceae* were found in the four samples. The amount in the Fayetteville sludge was low, being 1% in both the aerobic and anoxic tanks. However, the amount present in the Norman sludge was much greater, being 15.3% in the aerobic tank, and 7.4% in anoxic tank. These aerobic, gram-positive microorganisms are known to degrade atrazine.

Our results suggest that identifying the families of microorganism present in a given sludge will help determine the emerging contaminants that are likely to be biodegraded. As recycle and reuse of wastewater increases and treatment plants for direct potable reuse of wastewater are commissioned, it will be essential to determine the level of removal of emerging micropollutants. Further, for target micropollutants, for which removal depends on biodegradation, it will be important to periodically analyze the microorganism present in order to ensure the sludges continue to effectively biodegrade these compounds.

## Figures and Tables

**Figure 1 ijerph-17-02567-f001:**
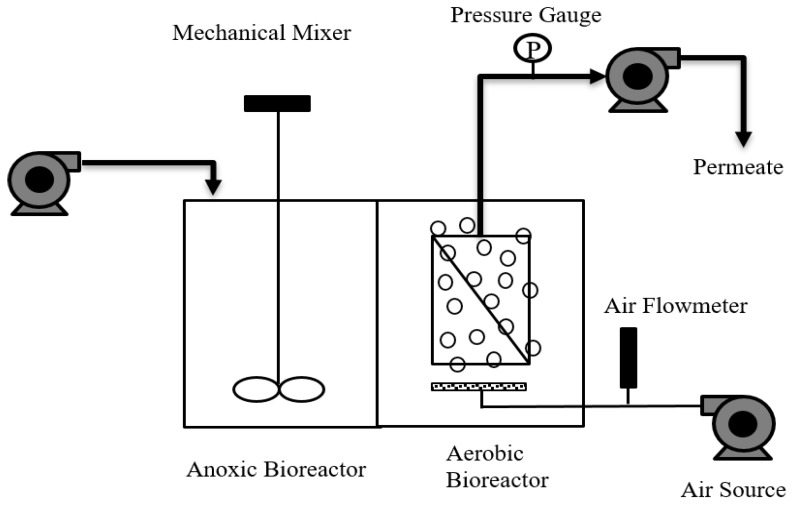
The laboratory-scale membrane bioreactor consisted of anoxic and aerobic membrane filtration tanks. Recirculation of the mixed liquor suspended solids between the anoxic and aerobic tanks ensured nitrification and denitrification at the two different redox potentials.

**Figure 2 ijerph-17-02567-f002:**
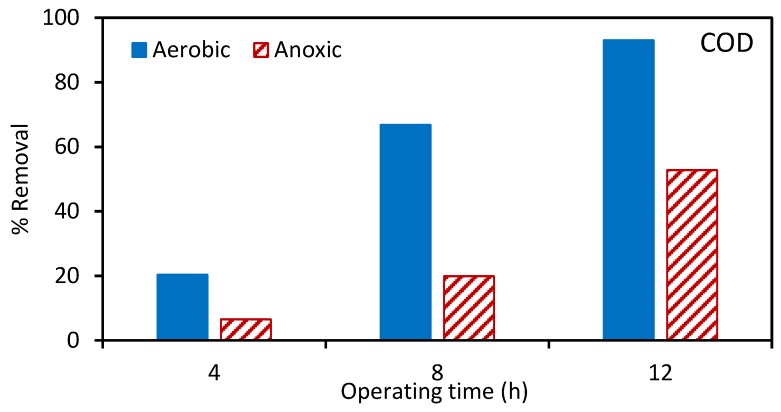
Percentage removal of chemical oxygen demand (COD) in the aerobic and anoxic tanks at 4, 8, and 12 h after spiking with atrazine. The total suspended solids (TSS) for the aerobic and anoxic tanks was 7029 and 6024 ± 600 mg/L, respectively.

**Figure 3 ijerph-17-02567-f003:**
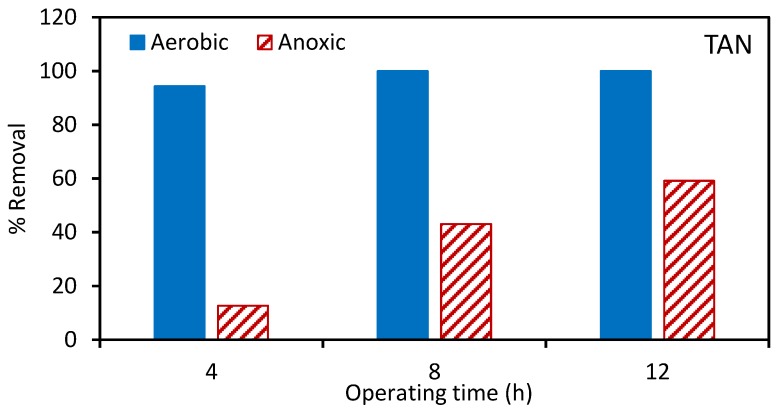
Percentage removal of total ammonia nitrogen (TAN) in the aerobic and anoxic tanks at 4, 8, and 12 h after spiking with atrazine. The total suspended solids (TSS) for the aerobic and anoxic tanks was 7029 and 6024 ± 600 mg/L, respectively.

**Figure 4 ijerph-17-02567-f004:**
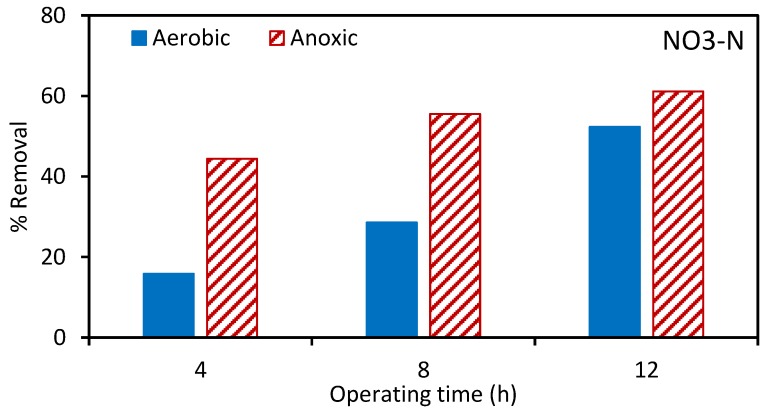
Percentage removal of nitrate nitrogen (NO_3_-N) in the aerobic and anoxic tanks at 4, 8, and 12 h after spiking with atrazine. The total suspended solids (TSS) for the aerobic and anoxic tanks was 7029 and 6024 ± 600 mg/L, respectively.

**Figure 5 ijerph-17-02567-f005:**
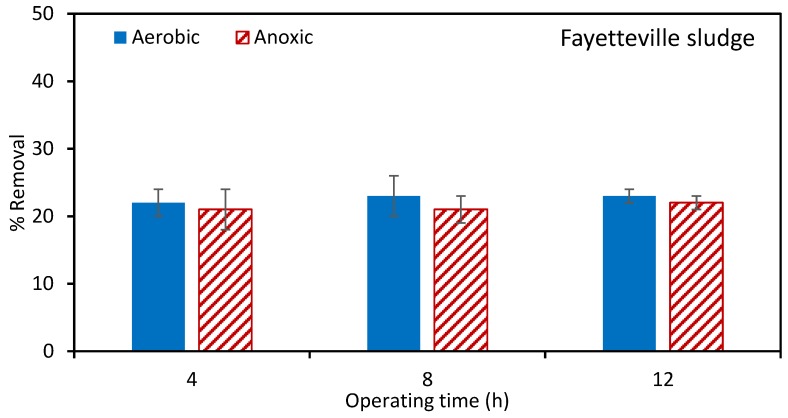
Percentage of atrazine removal using Fayetteville sludge. The initial atrazine concentration was 1 ppm. The total suspended solids (TSS) for aerobic and anoxic tanks was 6383 and 5023 ± 600 mg/L, respectively.

**Figure 6 ijerph-17-02567-f006:**
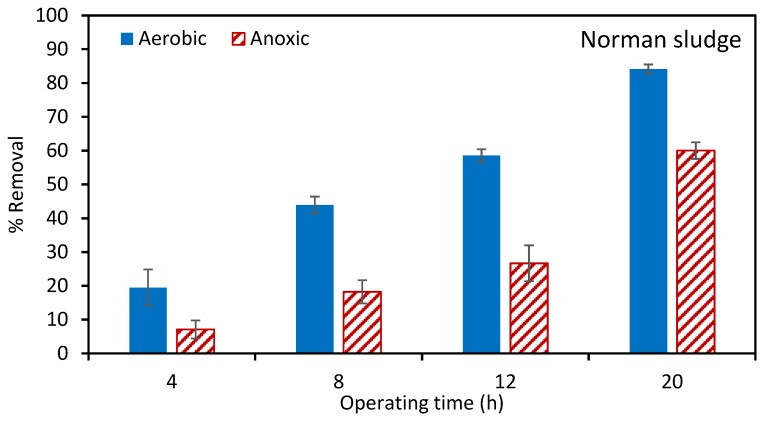
Parentage of atrazine removal using Norman sludge. The initial atrazine concentration was 1 ppm. The total suspended solids (TSS) for aerobic and anoxic tanks was 7244 and 5435 ± 600 mg/L, respectively.

**Figure 7 ijerph-17-02567-f007:**
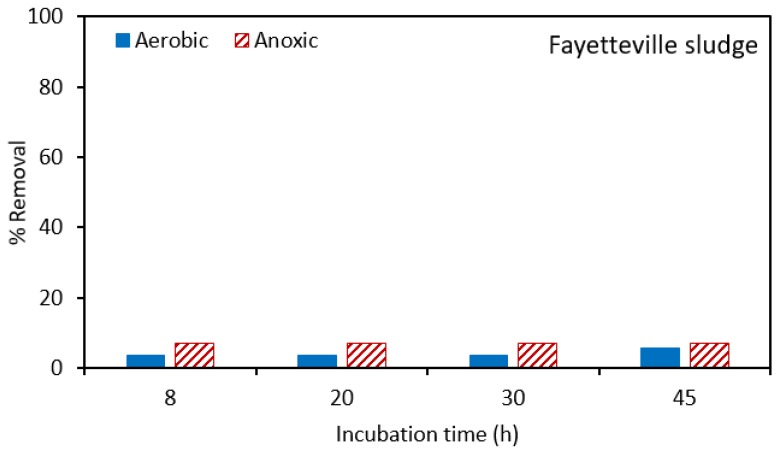
Percentage removal of atrazine in batch experiments where there is no mixing of the MLSS using Fayetteville sludge. The total suspended solids (TSS) in aerobic and anoxic tanks was 5958 and 4920 ± 600 mg/L, respectively.

**Figure 8 ijerph-17-02567-f008:**
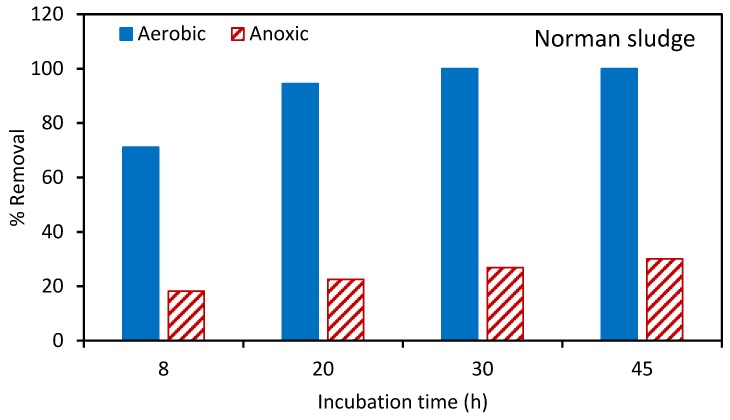
Percentage removal of atrazine in batch experiments where there is no mixing of the MLSS using Norman sludge. The total suspended solids (TSS) in aerobic and anoxic tanks was 7029 and 6024 ± 600 mg/L, respectively.

**Figure 9 ijerph-17-02567-f009:**
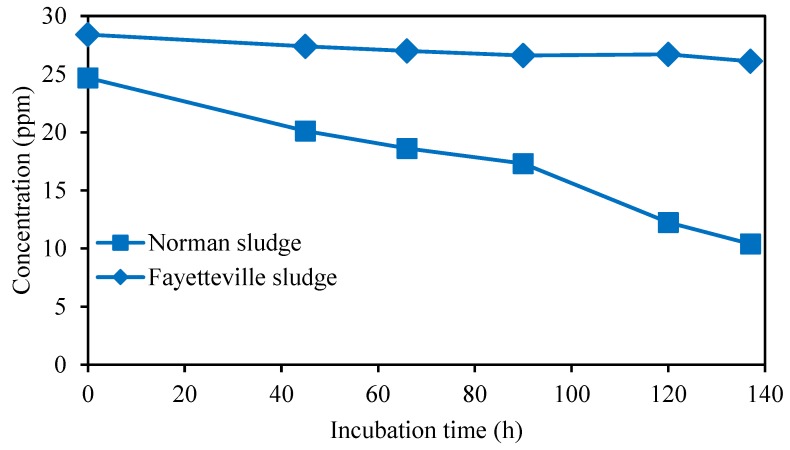
Decrease in atrazine concentration in Bushnell Hass broth as a function of time for Fayetteville and Norman sludges under aerobic conditions.

**Figure 10 ijerph-17-02567-f010:**
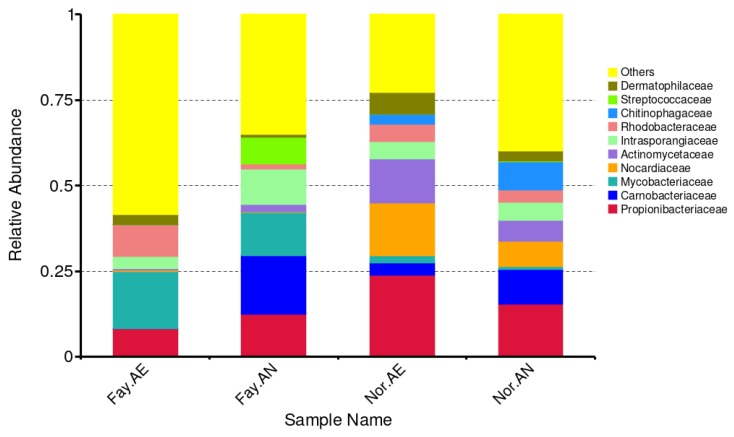
Taxonomy distribution at the family level for Fayetteville sludge from the aerobic (Fay-AE) and anoxic (Fay-AN) tanks and Norman sludge from the aerobic (Nor-AE) and anoxic (Nor-AN) tanks. The histogram shows only the abundance of the top 10 families. The other species are combined into “Others” in the Figure.

**Table 1 ijerph-17-02567-t001:** Composition of synthetic wastewater.

Compound	Concentration (mg/L)
Ammonium Acetate	240.88
monopotassium phosphate	43.94
Sodium Bicarbonate	125
Calcium chloride	10
Ferric Chloride	0.804
Yeast extract	50
Manganese sulfate	0.038
Zinc sulfate	0.035
Magnesium sulfate	25
Ferric chloride	0.375

**Table 2 ijerph-17-02567-t002:** Comparison of Bushnell Haas broth.

Compound	Concentration (mg/L)
Magnesium sulphate	200
Calcium chloride	20
Monopotassium phosphate	1000
Dipotassium phosphate	1000
Ammonium nitrate	1000
Ferric chloride	50
Final pH (at 25 °C)	7.0 ± 0.2

**Table 3 ijerph-17-02567-t003:** Variation in wastewater parameters for wastewater obtained from the West Side Wastewater Treatment Facility in Fayetteville over a 1 year period.

COD (mg/L)	TAN (mg/L)	NO_3_-N (mg/L)
1128–218	45–18	10.7–1.3

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
