# Peer review of "Atrazine Removal from Municipal Wastewater Using a Membrane Bioreactor"

_ijerph, 2020, doi:10.3390/ijerph17072567_

Round 1

Reviewer 1 Report

the topic is very interesting and very much necessary for the current world. Problem of potable water is increasing day by day so this manuscript will answer some of the questions . Some of the data does not have the standard deviation .So please add those. Conclusion should be rerite in respect to provide a take home message.

Author Response

Response to Reviewer 1

Question

the topic is very interesting and very much necessary for the current world. Problem of potable water is increasing day by day so this manuscript will answer some of the questions. Some of the data does not have the standard deviation. So please add those. Conclusion should be rewritten in respect to provide a take home message.

Answer

Since the number of samples is limited (usually 3-5) we have added ranges to the TSS values in Figures 2-8. 3-5 instead.  All readings are within ± 600 mg.

We have added the following to the conclusion.

Our results suggest that identifying the families of microorganism present in a given sludge will help determine the emerging contaminants that are likely to be biodegraded. As recycle and reuse of wastewater increases and treatment plants for direct potable reuse of wastewater are commissioned, it will be essential to determine the level of removal of emerging micropollutants. Further for target micropollutants, for which removal depends on biodegradation, it will be important to periodically analyze the microorganism present in order to ensure the sludges continues to effectively biodegrade these compounds.

Reviewer 2 Report

The paper addresses the removal of atrazine from wastewater using MBR and discovers some sludge that can seemingly achieve a high removal rate of atrazine in the MBR process. However, the description of the experimental procedure is not very detailed and the operation mode of MBR is also questionable. I would not recommend it for publication unless the authors address the concerns shown below:

(1) A major concern is associated with the operation of MBR. It is not clear whether the MBR has reached its steady state during the test. The authors used an SRT of 35 d, but how long did the authors operate the MBR? MBRs could be regarded to achieve their steady state after a long period of time of operation (such as 2 times SRT or even longer). Otherwise, the results might be misleading.

(2) For MBRs operated for a sufficient time to reach their steady state, the sludges may have quite different microbial communities from that of inoculums. The degradation ability of atrazine after long-term acclimation should be changed as well since microbes need a certain period of time to get them acclimated to the organic pollutants such as atrazine. Please take this into consideration to analyze and discuss your results.

(3) It is very strange that the MBR is operated in the semi-continuous mode and the whole design seems just like a batch reactor. The author did not provide any explanations why they used this mode for MBR operations since conventional MBR is mostly operated in a continuous mode.

(4) Under a semi-continuous mode, the removal rate will be quite different from the one obtained from the continuous mode, which raises the concern whether a high removal rate observed in this study can be achieved in real application. A continuous mode will be more convincing.

(5) What is the purpose of presenting Fig. 2,3,4 since they have nothing to do with the removal of atrazine? It would be better to put them in SI. Also, it is not indicated which sludge is used to get the results in these figures.

(6) Page 10, last paragraph, “After about 80 hours of incubation the rate of decrease of atrazine concentration increases suggesting that few other carbon sources are available to the microorganism.” The author should provide more explanations because the conclusion is very confusing.

(7) When did you collect the sludge samples from the reactors for analysis of microbial communities?

(8) There is no detailed information regarding the performance of membrane in this work. It would be nice if the performance if membrane is mentioned in the manuscript or SI.

(9) The evolution of biomass after inoculation is also important. However, in its current version, no detailed information of this is provided.

(10) The analysis and discussion of the microbial communities should be enhanced.

Reviewer 3 Report

The overall impression of the manuscript is strongly positive. It is an original research article that includes valuable experimental results.

The manuscript is about the removal of atrazine (contaminant of emerging concern) from municipal wastewater by MBR. Mainly, it is about differentiation of the removal mechanisms: adsorption vs biodegradation.

The manuscript is interesting, especially in the part of the next generation sequencing experiments. Although the results are not novel as authors admit in the conclusions, they are essential from the practical point of view. The manuscript has a sufficient impact on water reuse. It adds to the knowledge base on the removal of contaminants of emerging concern in water reuse schemes applying MBR.

The manuscript has sufficient literature review part in the introduction supported with 31 references. However, the authors could have used more references from the recent five years. The experiments are adequately designed and described in Materials & Methods. The obtained results are appropriately presented and discussed.

It is a well-written manuscript in grammatically correct American English.

The manuscript seems to adhere to the journal’s standards.

Specific comments:

  1. The difference between “the ability to introduce treated wastewater directly into the drinking water distribution system” and “direct potable reuse of wastewater” is not clear in the abstract (without reading the manuscript further). Do the authors mean indirect and direct potable water reuse? I understand that CEC secondary contamination risk is relevant for the indirect potable water reuse only, while for direct reuse trace contamination from WW is a risk (if exist for atrazine).
  2. “Increasing world population and urbanization has led not only to an increased need for drinking water but also increased discharge of wastewater” – the “increase” is mentioned 3 times, replace with synonyms.
  3. It is not clear in which meaning potable water reuse in an “extreme process”. Is it a social attitude that makes it extreme? Or because it is applied under extreme conditions (e.g. severe drought)?
  4. Do the US and the region of the study follow the WHO regulation for atrazine? Or is there a need to mention the US regulations and local limits (if not the same as WHO)?
  5. Regarding 2.3.: not sure if 35 L each tank or both, but maybe it becomes clear later. Better to specify at the beginning of 2.3.
  6. Probes that are later mentioned in the text can be shown in the Figure 1.
  7. Sludge recirculation can be shown in the Figure 1, if it occurs.
  8. Is the conclusion “Up to 20 % of the atrazine was removed by adsorption onto particulate matter present in the wastewater” adequately supported in the article? Or is it done based on the reference to the previous research only? Any desorption experiments conducted to prove the adsorption mechanism?
  9. May I suggest a title adjustment “Differentiation of Atrazine Removal Mechanisms in Municipal Wastewater Treatment with Membrane Bioreactor”

Round 2

Reviewer 2 Report

The revised manuscript has addressed my concerns, and it can be accepted for publication now.